# The Impact of Sidedness on the Efficacy of Anti-EGFR-Based First-Line Chemotherapy in Advanced Colorectal Cancer Patients in Real-Life Setting—A Nation-Wide Retrospective Analysis (RACER)

**DOI:** 10.3390/cancers15174361

**Published:** 2023-09-01

**Authors:** Paweł Michał Potocki, Rafał Wiśniowski, Dominik Haus, Zbyszko Chowaniec, Maciej Kozaczka, Magdalena Kustra, Marzenna Samborska-Plewicka, Marcin Szweda, Danuta Starzyczny-Słota, Magdalena Michalik, Grzegorz Słomian, Aneta Lebiedzińska, Natalia Jonak-Olczyk, Natalia Łaszewska-Kraińska, Krzysztof Adamowicz, Piotr Kolenda, Anna Drosik-Kwaśniewska, Marek Szwiec, Robert Dziura, Justyna Czech, Maria Dąbrowska, Ewa Nowakowska-Zajdel, Ewa Klank-Sokołowska, Kamil Konopka, Łukasz Kwinta, Jolanta Dobrzańska, Piotr J. Wysocki

**Affiliations:** 1Oncology Department, Faculty of Medicine, Jagiellonian University Medical College, 31-501 Cracow, Poland; 2Beskidzkie Centrum Onkologii, 43-300 Bielsko Biała, Poland; 3Oddział Onkologiczny, Wojewódzki Szpital Specjalistyczny, 59-220 Legnica, Poland; 4Department of Clinical Oncology, Lower Silesian Oncology Center, 53-413 Wroclaw, Poland; 52nd Clinic of Radiotherapy and Chemotherapy, Maria Sklodowska-Curie Memorial Cancer Center and National Institute of Oncology, Gliwice Branch, 44-102 Gliwice, Poland; 6Clinical and Experimental Oncology Department, Maria Skłodowska-Curie Memorial Cancer Center and National Institute of Oncology, Gliwice Branch, 44-102 Gliwice, Poland; 7Oncological Ward, Independent Public Health Care Unit, Voivodeship Specialized Hospital No. 3, 44-200 Rybnik, Poland; 8Department of Oncology, Collegium Medicum, University of Warmia and Mazury, 10-228 Olsztyn, Poland; 9Department of Oncology and Immuno-Oncology, Warmian-Masurian Cancer Center of the Ministry of the Interior and Administration Hospital, 10-228 Olsztyn, Poland; 10Oncology Clinic, University Hospital, 31-501 Cracow, Poland; 11Gdynia Oncology Centre of the Polish Red Cross Maritime Hospital, 81-519 Gdynia, Poland; 12Department of Clinical Oncology, Pomeranian Hospitals, 81-519 Gdynia, Poland; 13Department of Clinical Oncology and Immuno-Oncology, Greater Poland Cancer Centre, 61-866 Poznań, Poland; 14Department of Clinical Oncology, Maria Sklodowska-Curie Memorial Cancer Center and National Institute of Oncology, Cracow Branch, 31-115 Cracow, Poland; 15Department of Surgery and Oncology, University of Zielona Góra, 65-064 Zielona Góra, Poland; 16Department of Clinical Oncology, Holy Cross Cancer Center, 25-734 Kielce, Poland; 17Tadeusz Koszarowski Cancer Center in Opole, Department of Oncology with Daily Unit, 45-061 Opole, Poland; 18Department of Clinical Oncology, No. 4 Provincial Specialist Hospital, 41-902 Bytom, Poland; 19Department of Nutrition-Related Disease Prevention, Faculty of Health Sciences in Bytom, Medical University of Silesia in Katowice, 40-055 Katowice, Poland; 20Department of Metabolic Disease Prevention, Faculty of Health Sciences in Bytom, Medical University of Silesia in Katowice, 40-055 Katowice, Poland; 21Department of Clinical Oncology, Comprehensive Cancer Centre of Białystok, 15-027 Białystok, Poland

**Keywords:** anti-EGFR therapy, chemotherapy, colorectal cancer, cetuximab, panitumumab, FOLFIRI, FOLFOX, sidedness, Poland, real-world

## Abstract

**Simple Summary:**

Controlled clinical trials are one of the most important sources of medical knowledge, but their results are not always identical to those observed later in normal day-to-day practice. Therefore, real-world studies are needed to investigate those differences. The targeted agents cetuximab and panitumumab, combined with chemotherapy, are an effective treatment for metastatic colon cancer, provided that certain mutations have not occurred in cancer cells (namely activating mutations in *KRAS*, *NRAS* or *BRAF* genes). Recently, it was found that the outcomes of the aforementioned therapy differ depending on which part of the colon the tumour originates from. We conducted a real-world study on 842 patients treated at 16 cancer centres. The study confirmed that metastatic cancer that originated in the right part of the colon has a lower response to cetuximab or panitumumab-based treatment.

**Abstract:**

Anti-EGFR antibodies combined with chemotherapy doublets are a cornerstone of the upfront treatment of colorectal cancer. RAS and BRAF mutations are established negative predictive factors for such therapy. The primary tumour located in the proximal colon has recently emerged as another negative predictive factor. We have conducted a retrospective multicentre study to collect data on real-world population characteristics, practice patterns, and outcomes in patients with metastatic colorectal cancer treated in a first-line setting with either cetuximab or panitumumab in combination with either FOLFOX or FOLFIRI chemotherapy. The presented analysis focuses on the impact of the primary tumour location. 126 of 842 patients analysed (15.0%) had proximal primary. It was associated with a lower BMI at diagnosis, mucinous histology, and peritoneal metastases. It was also associated with inferior treatment outcomes in terms of response ratio: 59.4% vs. 74.22% (odds ratio [OR] 0.51, 95% CI 0.33–0.78, *p* = 0.010), and median depth of response: −36.7% vs. −50.0% (*p* = 0.038). There was only a borderline non-significant trend for inferior PFS in patients with proximal tumours. OS data was incomplete. The presented analysis confirms the negative impact of tumour sidedness on the efficacy of an upfront anti-EGFR-chemotherapy combination and provides valuable data on real-world population characteristics.

## 1. Introduction

Colorectal cancer (CRC) is the second most prevalent cancer worldwide and has the third highest mortality rate. In Poland, CRC is the second most common type of cancer with respect to prevalence and mortality [1,2]. According to estimates, 1 in 35 women and 1 in 22 men can develop CRC during their lifetime, and the number of new CRC cases per year is estimated to increase from 1.9 million in 2020 to 3.2 million in 2040 [3]. 

Metastatic disease (mCRC—metastatic colorectal cancer) that affects approximately half of CRC patients is associated with merely 10–15% of the 5-year survival rate [4,5]. Therefore, improving the efficacy of therapies for mCRC patients is of crucial importance. 

Currently, standard treatment for advanced disease involves cytotoxic chemotherapy combined with antiangiogenic drugs or antibodies targeting the epithelial growth factor receptor (EGFR). Immunotherapies that non-specifically stimulate the T-cell response, or other targeted drugs, are used relatively rarely [6].

The choice of first-line therapy is based primarily on the molecular profile of tumour tissue. The presence of activating mutations in genes encoding KRAS, NRAS, and BRAF kinases, as well as defects in DNA mismatch repair mechanisms, are the main predictive biomarkers driving clinical decisions. Alterations in other genes such as *PI3K*, *AKT*, *NTRK*, and *HER2* are currently of limited clinical utility. 

In the absence of *KRAS*, *NRAS*, or *BRAF* mutations, all major treatment guidelines recommend administration of a FOLFIRI (folate–5-fluorouracil–irinotecan) or FOLFOX (folate–5-fluorouracil–oxaliplatin) chemotherapy regimen combined with an anti-EGFR antibody—either cetuximab or panitumumab [6,7]. The addition of EGFR inhibitors significantly improves the therapeutic efficacy of chemotherapy alone in terms of overall survival (OS), progression-free survival (PFS), or objective responses (ORR) [8]. 

Although preselection according to *RAS* and *BRAF* status is now considered mandatory before the initiation of systemic treatment and the decision on the use of anti-EGFR agents, not all mutation-free tumours respond to therapy equally well. Therefore, several other predictive factors can be considered during initial treatment selection. Among those, tumour sidedness appears to be the most important, according to recent therapeutic guidelines. Colorectal tumours that arise from different parts of the intestine differ in more than one dimension. The proximal colon up to about halfway through the transverse colon (right side) originates from the embryonic midgut, and the remaining part (left side) is from the embryonic hindgut. The function and environmental exposure differ between the sides, and so does the vasculature. Cancers arising from the left side (LCC) or right side of the colon (RCC) differ in terms of their clinical, pathological and molecular characteristics, including prognosis and response to therapy. However, the exact molecular mechanisms driving the differences are not fully understood [9]. According to most studies, LCC is supposed to be associated with a better prognosis [10,11,12]. However, some investigations indicated a lack of difference in survival with respect to tumour sidedness or showed beneficial survival in patients with RCC compared to those with LCC [13,14,15]. A recent pooled analysis of several large clinical studies evaluating first-line systemic treatment based on a combination of chemotherapy and anti-EGFR agents (aEGFR-ChT) found that survival benefit is driven by patients with LCC [8,16]). Based on these findings, the latest European Society for Medical Oncology (ESMO) guidelines recommend first-line aEGFR-ChT only in patients with distal primaries (LCC) [6]. 

The purpose of this study was to characterise Polish patients treated with a first-line aEGFR-ChT agent within the national therapeutic programme with respect to practise patterns, treatment efficacy, and potential predictive factors. The aEGFR-ChT has been reimbursed in Poland since July 2017 under the Ministry of Health’s therapeutic programme [17]. Here, we present the study overview and first results on the effectiveness of therapy according to tumour sidedness in this population.

## 2. Materials and Methods

### 2.1. Study Design

RACER (Retrospective Analysis of Colorectal cancer, EGFR targeted therapy Results) is a retrospective real-world cohort study that was conducted in 16 oncological departments in Poland. It was based on medical data from consecutive mCRC patients treated with chemotherapy and an anti-EGFR combination within the national therapeutic programme (Figure 1). 

Physicians involved in the study collected individual patients’ data from medical records between February 2022 and February 2023. Data collected from medical records included demographic data; selected concomitant medications; the clinical, pathological, and molecular characteristics of cancer; the choice of first-line regimen; selected laboratory studies; response assessments; the duration of treatment and cause of discontinuation; patterns of progression; and the utilisation of local therapies for metastatic lesions. Analysis was planned according to tumour sidedness, regimen received, molecular testing methodology, and patterns of local interventions on metastases. 

The study was approved by the Bioethical Committee (23 February 2022 decision no 1072.61201.41.2022) with informed consent waived, as no direct patient participation was planned. 

### 2.2. National Therapeutic Programme B.4

The national Polish reimbursement policy restricts access to the most expensive therapies by creating Ministry of Health therapeutic programmes in which patients must be enrolled to receive treatment. Each programme has defined inclusion/exclusion criteria, allows for specified combination regimens, has a specific assessment schedule, and has discontinuation criteria. Adherence to the programmes is subject to an audit from regulatory authorities, with financial penalties applied for nonadherence. The reimbursement (within the B.4 programme) of FOLFIRI + cetuximab was introduced on 1 July 2017; of FOLFOX + panitumumab on 1 January 2018; of FOLFIRI + panitumumab on 1 November 2020; and of FOLFOX + cetuximab on 1 March 2021.

### 2.3. Patients

All patients in the analysis met the following inclusion criteria specified in the therapeutic programme (B.4):histologically confirmed colorectal cancer;metastatic disease (stage IV);disqualification from radical surgery;lack of prior systemic treatment due to metastatic disease;absence of mutations in the *KRAS* and *NRAS* genes (minimal requirements—evaluation of exons 2, 3 and 4 in both genes) and absence of the *BRAF V600E* mutation;disease assessable for response according to the RECIST 1.1 criteria;performance status 0–1 according to the Zubrod-WHO classification;over 18 years of age;results of the complete differentiated blood count:
platelet count ≥ 1.5 × 10^5^/mm^3^, an absolute neutrophil count ≥ 1500/mm^3^,haemoglobin ≥ 10.0 g/dL;adequate organ function:
total bilirubin concentration not exceeding 2 times the upper limit of normal (except for patients with Gilbert syndrome),serum transaminases (alanine and aspartic acid) activity not exceeding 5 times the upper limit of normal,creatinine concentration not exceeding 1.5 times the upper limit of normal;no contraindications to the FOLFIRI or FOLFOX chemotherapy regimen;exclusion of pregnancy;absence of brain metastases (in the case of clinical manifestations, exclusion based on imaging examination);no contraindications to cetuximab:
pulmonary fibrosis or interstitial pneumonia,hypersensitivity to any excipients.

The programme enforced discontinuation treatment in the event of: hypersensitivity to panitumumab, cetuximab, or any component of chemotherapy;disease progression;prolonged and clinically significant adverse events ≥ G3;pulmonary fibrosis or interstitial pneumonia;persistent deterioration of the performance status ECOG ≥ 3.

The choice of a particular systemic treatment regimen was made by the treating physician according to clinical judgment and the reimbursement options available at the time. Due to the lack of alternative reimbursement mechanisms, the scarcity of clinical trials at the time, and the high financial threshold for out-of-pocket financing, the investigators assumed that the analysed population was representative of all Polish mCRC patients undergoing first-line treatment with aEGFR-ChT at the time. The authors estimate that the study population comprises 7–10% of the total number of patients with mCRC treated with the anti-EGFR chemotherapy combination in Poland between 2017 and 2022. The overall number of patients treated with anti-EGFR agents in all lines of treatment in this period of time is reported to be 11,767, with no data available on usage in the first vs. subsequent lines (hence the patients enrolled in the study have represented no less than 7% of all treated patients) [18].

### 2.4. Evaluated Data

#### Characteristics of Patients

In this publication, demographic, clinical and pathological characteristics of patients were reported for the entire study population and according to primary tumour sidedness. According to the requirements of the national therapeutic programme, regular response assessments with CT scans were required at least every 3 months.

### 2.5. Endpoints

The following treatment effectiveness parameters were evaluated:

Progression-free survival (PFS)—defined as the time from the first administration of the investigated therapy to the investigator’s reported progression of the disease (PD) or death from any cause. When PD was not reported, the date of the last response evaluation entered was used instead, and the PFS had a censored status. 

The overall response rate (ORR) is defined as the percentage of patients who experience a partial or complete response to therapy. A dual-response assessment methodology was used. The ORR was based on the response assessment as stated in the medical records. Furthermore, when available, complete data for the RECIST 1.1 assessment were reported, including the dimensions of the target lesions and the sum of these dimensions. The response was then independently assessed based on these data, and the overall response rate by the investigator (ORRi) was calculated. Such an approach was assumed to mitigate the common practice of not using RECIST-based assessment methodology in real-world practice but instead only comparing the most current scan to the previous (and not baseline) one. 

Depth of response (DPR)—defined as the best percent change from baseline in the sum of longest diameter (SOD) for target lesions as reported by a local radiologist or, when unavailable, based on a set of target lesions retrospectively chosen (following RECIST 1.1 criteria) by the investigators having access to source imaging studies. Where neither the radiologist’s source nor retrospective assessment of the target lesion was possible, the status of DPR was marked as not assessable.

### 2.6. Statistical Analysis

Before analysis, data integrity was checked for logical and/or chronological errors at the central level. 

Statistical analysis was performed using the R environment [19]. Quantitative data were presented using basic descriptive statistics (mean SD, median, range), and qualitative data were presented as a percentage distribution of results. To test whether there are statistically significant differences between subgroups, the U Mann–Whitney test, *t*-test, chi-squared test, or Fisher exact test were used as needed. The level of statistical significance was assumed to be α = 0.05. 

## 3. Results

### 3.1. Characteristics of mCRC Patients Receiving Chemotherapy and Anti-EGFR Combination

A total of 842 patients (median age—64 years) who started treatment between 07.2017 and 11.2022 were investigated (Figure 1). The men accounted for two-thirds of the group. Patients with an increased body mass index (BMI) who were overweight or obese made up 59.6% of the study group, followed by those with a normal weight (38.1%). The median BMI was approximately 26 kg/m^2^, and the median body surface area (BSA) was 1.9 m^2^ (Table 1). 

Regarding disease characteristics, the vast majority of patients were diagnosed with tumour grade 2 (72.4%) and the absence of mucinous component (90.3%). Synchronous dissemination of the disease was observed in most patients (67.5%). The initial stage was characterised by the predominance of T3 scores (60.5%). At baseline, the most common site of metastasis was the liver (73.4%) (with the median size of the largest lesions being 44 mm), followed by extraregional lymph nodes (39.1%) and the lungs (28.4%). The least frequent location of metastases was bones (4.6%). Recurrence of primary tumour was observed in 34.1% of the cases. The baseline SOD ranged widely from 10 mm to 840 mm, with a median of 87.0 mm (Table 2 and Table 3).

Prior treatment for the primary tumour included mainly resection of the primary, which was performed in 70.1% of all patients and in 52.5% of patients with synchronous dissemination. Prior radiation therapy was used in 21.2% of the patients (almost all rectal cancers). Adjuvant/neoadjuvant chemotherapy was used in 34.5% of the patients and involved mainly fluoropirymidyne monotherapy. Only 11.2% of the patients underwent local therapy (resection, radiation therapy, ablation or embolisation) for oligometastatic disease prior to enrolment in the study. 

### 3.2. Characteristics of the Subgroups of Patients with Respect to Primary Tumour Location 

Among the 842 patients included in the study, three patients had primary lesions located both in the left and right colon, and data on sidedness was lacking for two patients. Therefore, the subgroups of patients with left-side (LCC) or right-side CRC (RCC) included 711 (84.4%) and 126 (15.0%) individuals, respectively. Patients from both groups did not differ in terms of sex and age (*p* < 0.05); however, LCC patients had a significantly higher BMI than RCC patients (median BMI 26.6 vs. 24.8 kg/m^2^, respectively, *p* = 0.022) (Table 1).

The most prevalent pathologic stage of primary lesion in both groups was pT3, and no clear difference in regional lymph node involvement was noted (66.7% in RCC and 67.8% in LCC patients). For patients with metachronous metastases, there was a trend towards a lower stage of primary tumour in RCC patients. The RCC patients presented more often with high-grade (G3) tumours, and the LCC patients had more moderate-grade (G2) tumours; however, the differences were not statistically significant (*p* = 0.115). Mucinous component of tumours was detected more frequently in RCC patients compared to LCC (24.2% vs. 7.0%, *p* < 0.001). In both groups, the majority of patients were diagnosed with synchronically disseminated disease.

Peritoneal metastases occurred significantly more frequently in RCC patients (43.7% vs. 18.9%, *p* < 0.001). In contrast, liver, lung or bone metastases were more common in the LCC group, but this was not statistically significant (*p* > 0.05). Locoregional recurrence of the primary tumour was observed significantly more frequently in LCC patients than in RCC patients (36.0% vs. 22.0%, *p* = 0.004). The baseline median values of SOD were very similar in both groups (about 87 mm). 

The groups differed in terms of prior treatment. Primary tumour resection was performed more often in RCC patients (84.0% vs. 67.7%, *p* < 0.001), and radiotherapy was applied more often in LCC patients (24.4% vs. 4.0%, *p* < 0.001), which was obviously driven by rectal cancers. Adjuvant/neoadjuvant chemotherapy was used in a similar proportion in both populations (34.5%), but capecitabine alone was more frequently administered in LCC patients (24.3% vs. 15.9%, *p* = 0.012) and capecitabine/oxaliplatin combination in RCC patients (18.6% vs. 10.2%). 

### 3.3. The Impact of Sidedness on the Efficacy of Systemic Treatment

#### 3.3.1. Tumour Response

At the data cutoff of 15.01.2023, the median follow-up was 329 days (interquartile range [IQR] 165–565). A total of 631 subjects were eligible for the response analysis. The overall response rate (complete or partial response) as assessed by the investigators (ORR) was 71.8% (95% CI; 68.1 to 75.2). The ORR in LCC patients (74.22%; 95% CI; 70.2 to 77.9%) was significantly higher than in RCC patients (59.4%; 95% CI, 49.2 to 68.9%)—the odds ratio (OR) for objective response was 1.97 (95% CI; 1.27–3.01, *p* = 0.010) (Figure 2). 

Complete response was achieved in 9.3% vs. 11.9% (OR 0.76, 95% CI 0.39–1.48) and partial response in 65.0% and 47.5% (OR 2.01, 95% CI 1.31–3.09) of LCC and RCC patients, respectively. Clinical benefit (CR + PR + SD) was achieved in 93.5% and 90.1% (OR 1.97, 95% CI 1.271–3.07) of LCC and RCC patients, respectively (Table 4).

#### 3.3.2. Depth of Response

The median DPR in the overall population was −41.9%. The DPR was significantly higher in LCC patients: −50.0% (IQR: −69.6% to −23.8%) than in RCC patients: −36.7% (IQR: −68.5% to −8.1%), *p* = 0.038 (Table 4, Figure 3). 

#### 3.3.3. Progression-Free Survival

A total of 756 patients were eligible for the PFS analysis, with 456 of them having experienced progression or death at the data cutoff. The median PFS was 346 days (95% CI, 332 to 378 days) for the overall population, 354 days (95% CI, 338 to 392 days) for the LCC and 289 days (95% CI, 256 to 382 days) for the RCC subpopulation. The differences were borderline non-significant (*p* = 0.51) (Figure 4). Overall survival data were incomplete at the time of the analysis, with data on survival available for only 173 subjects.

## 4. Discussion

This is the first study investigating the results of first-line anti-EGFR-based therapy in the real-world population of Polish mCRC patients. 

The characteristics of the population generally followed the patterns of previously published analyses [20,21]; however, the prevalence of RCC turned out to be lower than expected. In the largest and most recent analysis of the impact of tumour sidedness on treatment outcomes, patients with RCC represented 25.5% of the entire mCRC population [8], compared to 15% in our study. On the contrary, in two recently published cross-sectional studies of consecutive Polish CRC patients, the prevalence of RCC was 14.7% [22] or 11.9% [23] in the subgroup without the *RAS* and *BRAF* mutations, which is similar to the value reported in this study. Therefore, the lower prevalence of RCC compared to the data from large clinical trials seems to be typical of the Polish population. The reason for this difference remains unclear. Selection bias may be a contributing factor, as patients with more aggressive RCCs might have had a lower chance of starting treatment and being included in the study. The association of lower BMI with RCC presumably reflects a higher incidence of cachexia characteristic of more aggressive neoplasms such as RCC. This may also be a pathomechanism contributing to the difference in treatment results observed between RCC and LCC and requires further investigation. A higher prevalence of mucinous or partially mucinous histology in patients with RCC is consistent with previously published studies [21,22]. Patients with RCC presented more often with metastatic disease, reflecting its more aggressive characteristics. The stage distribution was otherwise similar between subgroups, especially with respect to nodal involvement. Similarly, the prior use of adjuvant chemotherapy was similar between subgroups. A significantly higher utilisation of oxaliplatin-based regimens in RCC is probably related to the differences in the initial stages of patients who were treated radically prior to recurrence and study entry. Stage distribution difference, although not significant, appears to reflect a higher risk of recurrence associated with stage II RCC than with stage II LCC. 

Our study confirmed that LCC patients treated with upfront anti-EGFR-CHT achieved superior outcomes (ORR and DPR) compared to RCC patients, which is consistent with the literature and reflects previously described differences in the clinical course of cancers arising from different parts of the colon [8]. There was also a trend towards better PFS in left-sided tumours, but the difference has nearly missed statistical significance. This observation was not consistent with previously published data, which show a strong negative correlation between RCC and PFS [8,16]. Several possible factors could influence the results of our analysis. The reported rate of primary right-side tumours in the studied population of mCRC patients was among the lowest reported for studies evaluating first-line anti-EGFR-based therapy. At the data cut-off, among the 846 subjects in the database, a PFS event was reported for 456 patients, and data were missing for 90 patients. This is probably due to the retrospective nature of our analysis. Another possible explanation is the discrepancy in the quality of molecular diagnostics. Since in Poland many laboratories testing for the *RAS/BRAF* mutation do not undergo periodic external quality control assessment, a worrying discrepancy in the sensitivity of genetic assessment between various laboratories has been observed [24,25]. The differences in the number of false negatives (patients considered ‘wild type’ with the mutation undetected) in cases of LCC versus RCC may therefore have contributed to the observed results. The possible causes of the discrepancy will be explored in a future study.

The authors are aware of the weaknesses of the study (retrospective, academic analysis); therefore, they have made an extensive effort to mitigate them. The study was not a prospective clinical trial, and the patient’s data have only been internally validated. Due to the retrospective nature of the study, the data on a number of potential prognostic factors were not complete in time to be included in this publication and could therefore not be included in multivariate analysis (most notably the treatment received and dosing intensity). The authors plan to continue the data collection and publish updated results, including the missing variables. It should also be noted that the national reimbursement policy in Poland positively contributed to the homogeneity of the population as inclusion/exclusion criteria, treatment regimens and response assessments were predefined and closely monitored during treatment. However, these same factors might have been a source of selection bias, negatively impacting the ability to extrapolate these data to different populations. 

Furthermore, due to the retrospective nature of the study, some inaccuracies in the database may have occurred. Therefore, prior to statistical analysis, data curation was performed to check for logical and chronological errors. We believe that such management, together with the approach described in the methodology section, increased the reliability of the results of our research.

## 5. Conclusions

This large retrospective analysis that evaluated a homogeneous population of mCRC patients undergoing EGFR-based first-line chemotherapy demonstrated that patients with left-sided primary tumours experienced significantly better outcomes in terms of ORR and DPR. It also provided the first data on real-world treatment patterns in this setting in Poland. 

## Figures and Tables

**Figure 1 cancers-15-04361-f001:**
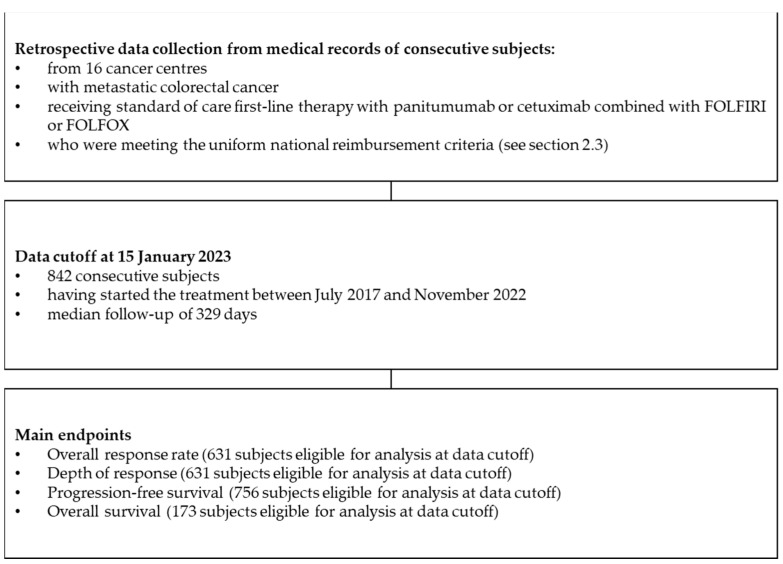
Flowchart illustrating the study design.

**Figure 2 cancers-15-04361-f002:**
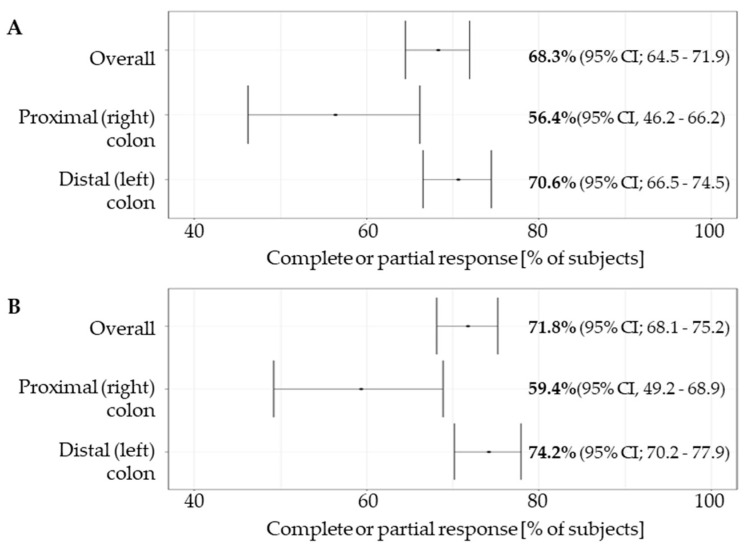
Overall response rate (**A**) and investigator assessed overall response rate (**B**) in the overall study group and in subgroups based on the tumour sidedness. Overall response rate was reported as per standard of care medical records. Overall response rate by investigator was independently assessed based on independent measurement of lesions by the investigators extracting the data (see Section 2.5).

**Figure 3 cancers-15-04361-f003:**
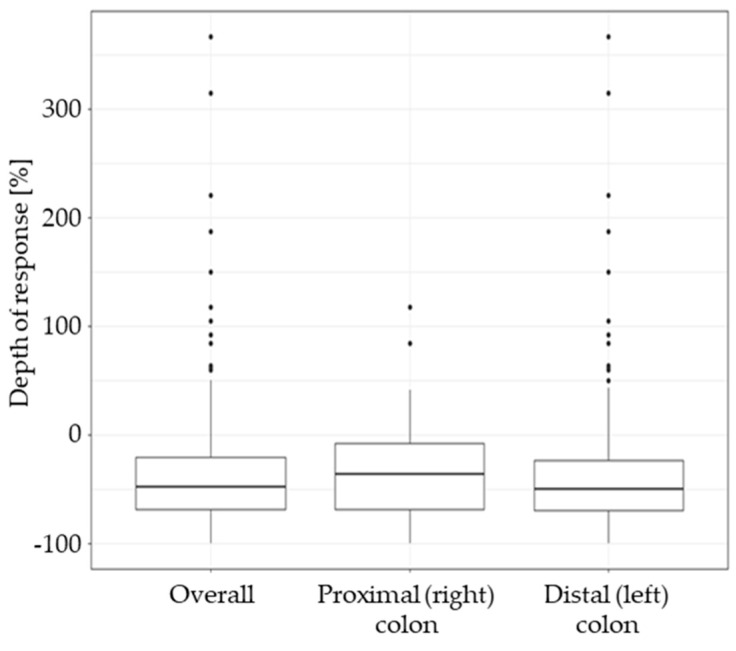
Depth of response (DPR) in the overall study group and in subgroups based on the tumour sidedness. The DPR is expressed as the percent change of baseline sum of lesion diameters (negative values indicate tumour shrinkage, positive values indicate tumour growth).

**Figure 4 cancers-15-04361-f004:**
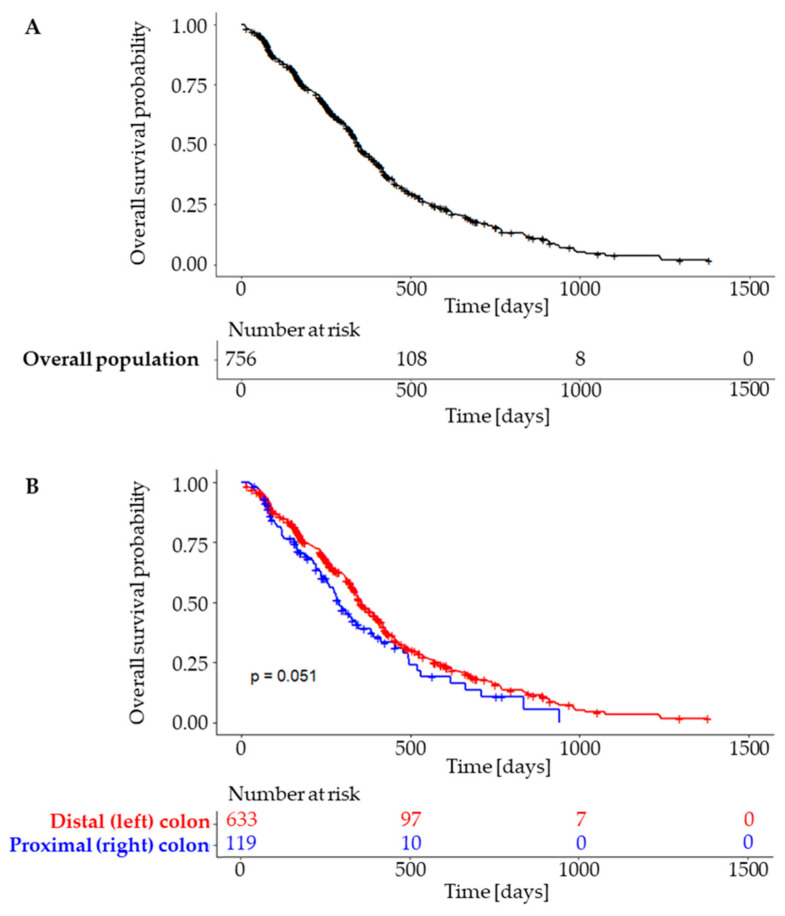
Progression free survival in overall population (**A**) and subgroups according to tumour sidedness (**B**).

**Table 1 cancers-15-04361-t001:** Demographic characteristics of patients in overall study group and in subgroups based on sidedness of the tumour.

Parameter Category	Overall, N = 842	Right Colon N = 126	Left Colon, N = 711	Right Colon vs. Left Colon
Sex, N (%)				0.287 #
Male	559 (66.4%)	78 (61.9%)	478 (67.2%)	
Female	283 (33.6%)	48 (38.1%)	233 (32.8%)	
Age [years]				0.128 ^
N	842	126	711	
Mean (SD)	62.3 (9.8)	63.5 (9.4)	62.1 (9.9)	
Median (Q1; Q3)	64.0 (57.0; 70.0)	65.0 (58.0; 70.0)	63.0 (57.0; 69.0)	
Min; max	25.0; 83.0	35.0; 80.0	25.0; 83.0	
Weight [kg]				0.058 ^
N	826	126	698	
Mean (SD)	76.3 (16.7)	74.0 (17.3)	76.8 (16.6)	
Median (Q1; Q3)	75.0 (64.0; 87.0)	72.0 (60.0; 85.0)	76.0 (64.0; 87.0)	
Min; max	39.5; 140.0	40.0; 128.0	39.5; 140.0	
Height [cm]				0.869 ^
N	827	126	699	
Mean (SD)	168.8 (9.4)	169.1 (10.0)	168.8 (9.3)	
Median (Q1; Q3)	169.0 (163.0; 176.0)	170.0 (162.2; 176.0)	169.0 (163.0; 175.0)	
Min; max	125.0; 198.0	144.0; 197.0	125.0; 198.0	
BMI [kg/m^2^]				0.007 ^
N	796	124	670	
Mean (SD)	26.8 (5.0)	25.7 (4.9)	27.0 (5.1)	
Median (Q1; Q3)	26.3 (23.3; 29.8)	24.8 (22.3; 28.4)	26.6 (23.4; 30.0)	
Min; max	15.6; 52.3	15.6; 43.8	15.6; 52.3	
BMI—category, N (%)				0.022 &
Underweight	19 (2.4%)	5 (4.0%)	14 (2.1%)	
Normal	303 (38.1%)	60 (48.4%)	242 (36.1%)	
Overweight	280 (35.2%)	35 (28.2%)	244 (36.4%)	
Obese	194 (24.4%)	24 (19.4%)	170 (25.4%)	
BSA [m^2^]				0.207 ^
N	796	124	670	
Mean (SD)	1.9 (0.2)	1.8 (0.2)	1.9 (0.2)	
Median (Q1; Q3)	1.9 (1.7; 2.0)	1.8 (1.6; 2.0)	1.9 (1.7; 2.0)	
Min; max	1.3; 2.5	1.4; 2.5	1.3; 2.5	

BMI, body mass index; BSA, body surface area; cm, centimetre; kg, kilogram; m, meter; N, number, SD, standard deviation, Q, quartile. ^—U Mann–Whitney test, #—chi-squared test, &—Fisher exact test.

**Table 2 cancers-15-04361-t002:** Disease characteristics in overall study group and in subgroups based on sidedness of the tumour.

Parameter Category	Overall, N = 842	Right Colon N = 126	Left Colon, N = 711	Right Colon vs. Left Colon
Tumour grade, N (%)				0.115 #
1	92 (12.8%)	12 (10.5%)	80 (13.2%)	
2	522 (72.4%)	78 (68.4%)	443 (73.1%)	
3	107 (14.8%)	24 (21.1%)	83 (13.7%)	
Mucinous component, N (%)				<0.001 #
Present	76 (9.7%)	30 (24.2%)	46 (7.0%)	
Absent	709 (90.3%)	94 (75.8%)	615 (93.0%)	
Initial T score, N (%) (all patients)				0.177 &
Tx	94 (12.0%)	12 (9.8%)	82 (12.4%)	
T1	4 (0.5%)	1 (0.8%)	3 (0.5%)	
T2	49 (6.3%)	6 (4.9%)	43 (6.5%)	
T3	474 (60.5%)	70 (57.4%)	404 (61.1%)	
T4	72 (9.2%)	10 (8.2%)	62 (9.4%)	
T4a	38 (4.9%)	11 (9.0%)	27 (4.1%)	
T4b	52 (6.6%)	12 (9.8%)	40 (6.1%)	
Initial N score, N (%) (all patients)				0.824 &
Nx	125 (16.0%)	17 (14.2%)	108 (16.3%)	
N0	128 (16.4%)	23 (19.2%)	105 (15.9%)	
N1	74 (9.5%)	7 (5.8%)	67 (10.1%)	
N1a	49 (6.3%)	7 (5.8%)	42 (6.4%)	
N1b	89 (11.4%)	16 (13.3%)	73 (11.0%)	
N1c	25 (3.2%)	3 (2.5%)	22 (3.3%)	
N2	83 (10.6%)	13 (10.8%)	70 (10.6%)	
N2a	92 (11.8%)	17 (14.2%)	75 (11.3%)	
N2b	116 (14.9%)	17 (14.2%)	99 (15.0%)	
Metastases occurrence, N (%)				<0.001 #
Synchronous	560 (67.5%)	93 (73.8%)	466 (66.4%)	
Metachronous	270 (32.5%)	33 (26.2%)	236 (33.6%)	
Initial stage for metachronous patients, N (%)				0.071 #
I	14 (5.5%)	0 (0%)	14 (6.4%)	
II	57 (22.4%)	12 (35.3%)	45 (20.5%)	
III	183 (72%)	22 (64.7%)	160 (73.1%)	
Prior resection of the primary tumour, N (%)				<0.001 #
Yes	558 (70.1%)	100 (84.0%)	457 (67.7%)	
No	238 (29.9%)	19 (16.0%)	218 (32.3%)	
Prior radiotherapy to the primary tumour, N (%)				<0.001 #
Yes	178 (21.2%)	5 (4.0%)	173 (24.4%)	
No	660 (78.8%)	121 (96.0%)	537 (75.6%)	
Prior adjuvant/neoadjuvant chemotherapy, N (%)				0.012 #
Fluoropirymidine	173 (23.0%)	18 (15.9%)	155 (24.3%)	
Fluoropirymidine + oxaliplatin	86 (11.5%)	21 (18.6%)	65 (10.2%)	
No	492 (65.5%)	74 (65.5%)	417 (65.5%)	
Prior localized therapy for oligometastatic disease, N (%)				0.290 #
Yes	93 (11.2%)	18 (14.4%)	75 (10.7%)	
No	735 (88.8%)	107 (85.6%)	627 (89.3%)	
Sample origin for RAS/BRAF testing, N (%)				0.079 &
Primary tumour	782 (94.0%)	114 (90.5%)	667 (94.6%)	
Liver metastasis	28 (3.4%)	5 (4.0%)	23 (3.3%)	
Non-liver metastasis	22 (2.6%)	7 (5.6%)	15 (2.1%)	
Dissemination, N (%)				0.125 ^
Synchronous	560 (67.5%)	93 (73.8%)	466 (66.4%)	
Metachronous	270 (32.5%)	33 (26.2%)	236 (33.6%)	

N, number. ^—U Mann–Whitney test, #—chi-squared test, &—Fisher exact test.

**Table 3 cancers-15-04361-t003:** Occurrence of metastasis in overall study group and in subgroups based on sidedness of the tumour.

Parameter Category	Overall, N = 842	Right Colon, N = 126	Left Colon, N = 711	Right Colon vs. Left Colon
Metastasis to the liver, N (%)				0.053 #
Yes	615 (73.4%)	83 (65.9%)	529 (74.6%)	
No	223 (26.6%)	43 (34.1%)	180 (25.4%)	
Metastasis to the lungs, N (%)				0.081 #
Yes	237 (28.4%)	27 (21.6%)	210 (29.7%)	
No	598 (71.6%)	98 (78.4%)	497 (70.3%)	
Metastasis to extra-regional lymph nodes, N (%)				0.095 #
Yes	328 (39.1%)	58 (46.4%)	270 (38.0%)	
No	510 (60.9%)	67 (53.6%)	440 (62.0%)	
Metastasis to bones, N (%)				0.862 #
Yes	39 (4.6%)	5 (4.0%)	34 (4.8%)	
No	800 (95.4%)	121 (96.0%)	676 (95.2%)	
Metastases in the peritoneal cavity, N (%)				<0.001 #
Yes	189 (22.6%)	55 (43.7%)	134 (18.9%)	
No	647 (77.4%)	71 (56.3%)	574 (81.1%)	
Primary tumour/local recurrence, N (%)				0.004 #
Yes	267 (34.1%)	26 (22.0%)	239 (36.0%)	
No	517 (65.9%)	92 (78.0%)	425 (64.0%)	
Metastasis to other locations, N (%)				0.226 #
Yes	90 (10.8%)	18 (14.3%)	72 (10.2%)	
No	746 (89.2%)	108 (85.7%)	635 (89.8%)	
Baseline SOD (mm)				0.319 ^
N	759	112	643	
mean (SD)	105.5 (77.9)	95.3 (60.4)	107.5 (80.6)	
median (Q1; Q3)	87.0 (51.0; 139.0)	87.5 (41.8; 130.8)	87.0 (51.0; 140.0)	
min; max	10.0; 840.0	10.0; 306.0	12.0; 840.0	
Largest dimension of the largest of the liver lesions (mm)				0.212 ^
N	598	79	515	
mean (SD)	52.8 (35.3)	48.5 (32.2)	53.6 (35.7)	
median (Q1; Q3)	44.0 (26.2; 70.0)	40.0 (24.5; 68.5)	45.0 (27.0; 70.0)	
min; max	1.9; 235.0	8.0; 144.0	1.9; 235.0	

Mm, millimetre; N, number; SD, standard deviation; SOD, sum of diameters; Q, quartile, ^—U Mann–Whitney test, #—chi-squared test.

**Table 4 cancers-15-04361-t004:** Response rate and depth of response according to tumour sidedness.

Parameter Category	Overall, N = 631	Proximal (Right) Colon, N = 101	Distal (Left) Colon, N = 528	Proximal (Right) Colon vs. Distal (Left) Colon
Depth of response [%]				0.038 ^
N	631	101	528	
Mean (SD)	−41.9 (45.1)	−36.7 (41.6)	−43.0 (45.7)	
Median (Q1; Q3)	−47.9 (−68.9; −20.2)	−36.7 (−68.5; −8.1)	−50.0 (−69.6; −23.8)	
Min; max	−100.0; 366.7	−100.0; 117.4	−100.0; 366.7	
Overall response rate, N (%)				0.010 #
Complete response	55 (8.7%)	12 (11.9%)	43 (8.1%)	
Partial response	376 (59.6%)	45 (44.6%)	330 (62.5%)	
Stable disease	168 (26.6%)	37 (36.6%)	130 (24.6%)	
Progressive disease	32 (5.1%)	7 (6.9%)	25 (4.7%)	
Complete or partial response	431 (68.3%)	57 (56.4%)	373 (70.6%)	−14.2 (−25.2; −3.2)
95% confidence interval	64.5; 71.9	46.2; 66.2	66.5; 74.5	
Overall response rate by investigator, N (%)				0.010 #
Complete response	61 (9.7%)	12 (11.9%)	49 (9.3%)	
Partial response	392 (62.1%)	48 (47.5%)	343 (65.0%)	
Stable disease	134 (21.2%)	31 (30.7%)	102 (19.3%)	
Progressive disease	44 (7.0%)	10 (9.9%)	34 (6.4%)	
Complete or partial response	453 (71.8%)	60 (59.4%)	392 (74.2%)	−14.8 (−25.7; −4.0)
95% confidence interval	68.1; 75.2	49.2; 68.9	70.2; 77.9	

^—U Mann–Whitney test, #—chi-squared test. Overall response rate was reported as per standard of care medical records. Overall response rate by investigator was independently assessed based on independent measurement of lesions by the investigators extracting the data (see Section 2.5).

## Data Availability

The data presented in this study are available upon request from the corresponding author. The data are not publicly available due to ethical and legal concerns.

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
