# Peer review of "The Impact of Sidedness on the Efficacy of Anti-EGFR-Based First-Line Chemotherapy in Advanced Colorectal Cancer Patients in Real-Life Setting—A Nation-Wide Retrospective Analysis (RACER)"

_cancers, 2023, doi:10.3390/cancers15174361_

Round 1
Reviewer 1 Report
The manuscript delves into a compelling topic regarding the influence of sidedness on the effectiveness of anti-EGFR-based first-line chemotherapy in advanced colorectal cancer patients. The study's design and analysis are commendable, providing valuable insights into this area of research.
Minor Comments:
The manuscript could benefit from a clearer flowchart illustrating the study design. This addition would enhance the reader's understanding of the research process.
On page 12, Figure 1 requires revision. Reducing the bar and plot size while increasing text size and resolution would significantly improve its visual clarity.
A figure (Figure 2) seems to be missing.
For Figure 3 on page 14, please consider adding labels "A" and "B" to each panel. Additionally, increasing the text size in the figure would enhance its legibility and impact.
Line 342-343, it is recommended to include relevant references to support the statements made.
Line 345 would benefit from the addition of supporting references
Look good. Minor editing of English language required
Author Response
Respected Colleague,
Thank you for the review, for the time you took assessing our manuscript, and for all the valuable comments regarding the manuscript’s shortcomings. We have addressed the outlined issues in a following manner:
- The manuscript could benefit from a clearer flowchart illustrating the study design. This addition would enhance the reader's understanding of the research process.
- A flowchart has been added for clearer depiction of study design (Figure 1.).
- On page 12, Figure 1 requires revision. Reducing the bar and plot size while increasing text size and resolution would significantly improve its visual clarity.
- Revised the figure in question (Figure 2. In a revised version of the manuscript) to address the issues noticed by the reviewer. Similar changes were made to the remaining figures. A second methodology of ORR assessment was added and the caption was reworked following the remark by the other reviewer.
- A figure (Figure 2) seems to be missing.
- A fluke in text formatting. Thank you for noticing. The Figure was re-added (now as Fig. 3) with modifications similar to suggested in the previous point.
- For Figure 3 on page 14, please consider adding labels "A" and "B" to each panel. Additionally, increasing the text size in the figure would enhance its legibility and impact.
- The Figure in question (now Figure 4) has been reworked.
- Line 342-343, it is recommended to include relevant references to support the statements made.
- The sentence (lines 364-365 in the revised manuscript) has been rephrased. At the time of manuscript preparation we were aware (through unpublished sources) of one similar project being undertaken but we struggled to find the publication. We have since confirmed with the authors of the study in question that it has not been published yet – hance the change of statement. Additionally a passage on the study being the first source of national RWD data was added to the conclusions (lines 432 – 433).
- Line 345 would benefit from the addition of supporting references
- The claim on the population characteristics being comparable to previously published studies ( now in line 367) has been supplemented with references. The claim about lower than expected prevalence of RCC is explained and referenced in the next sentence.
We hope you will find the changes to the manuscript acceptable.
Thank you again.
Sincerely,
The Authors.

Reviewer 2 Report
In this retrospective multicentre study, the authors have evaluated the impact of sidedness in patients with metastatic colorectal cancer receiving a chemotherapy regimen combined with an anti-EGFR antibody (either cetuximab or panitumumab). With 842 patients from 16 oncological departments in Poland, they found that patients with left-side primary tumour achieved significantly better outcomes in terms of overall response and depth of response. According to the authors, this study constitutes the largest one in Poland.
In my opinion, this work is clinical in nature but fits the scope of Cancers journal. However, it has some limitations and several changes should be performed prior publication:
· Line 43: “provided that certain mutations have not occurred in cancer cells”. You should better explain this statement.
· Line 53: “untreated metastatic colorectal cancer, treated with”. You should better explain this statement: untreated or treated?
· Line 59: “and median progression-free survival 289 vs 354 days although the last difference was borderline non-significant (p=0.51)”. Non-significant differences should not be included in the abstract.
· Line 134: The study was approved by the Bioethical Committee (23.02.2022 decision no 134 1072.61201.41.2022) with informed consent waived, as no direct patient participation was planned. Is it OK for the journal?
· Line 242-244: Please define abbreviations BMI and BSA when firstly used in the text.
· Line 258: Please define abbreviations SOD when firstly used in the text.
· Significant differences in BMI between LCC and RCC are shown in Table 1, but it is not stated in the text.
· Line 295-296: Locoregional recurrence of the primary tumour was observed more frequently in LCC than RCC patients (36.0% vs 296 22.0%). Are these differences significant? Please specify.
· The differences between “Overall Response Rate” and “Overall Response Rate by Investigator” are not clearly stated in the text. Please include a brief explanation or omit one of them.
· Line 315-318: Statistical analysis with p-values should be mentioned in the text when significant.
· Figure 2 about depth of response is missing.
· Line 342-343: “All other previously published studies are scarce and single-centre”. Please include these literature references and compare/discuss their results with yours.
· The authors should expand the limitations in Discussion section: selection bias, residual confounding (as relevant covariates like different treatments are not assessed), results extrapolation to other populations might be compromised, etc.
· Response to the treatment could be influenced by body mass index instead of the sidedness of the tumour? In line 281, authors state that “LCC patients had significantly higher BMI than RCC patients”. It would be interesting to include your thoughts about that in Discussion section.
· Line 401: “with a trend towards improvement in PFS”. Please avoid to state non significant differences in Conclusion section.
Author Response
Respected Colleague,
Thank you for the review, for the time you took assessing our manuscript, and for all the valuable comments regarding the manuscript’s shortcomings. We have addressed the outlined issues in a following manner:
- Line 43: “provided that certain mutations have not occurred in cancer cells”. You should better explain this statement.
- The mutations in question were named in parentheses (line 43-44 in the revised manuscript).
- Line 53: “untreated metastatic colorectal cancer, treated with”. You should better explain this statement: untreated or treated?
- This indeed sounds unclear. Thank you for noticing. We have rephrased this sentence in the abstract to avoid phrases that may be misunderstood (lines 53-55 in the revised manuscript).
- Line 59: “and median progression-free survival 289 vs 354 days although the last difference was borderline non-significant (p=0.51)”. Non-significant differences should not be included in the abstract.
- The line in question was shortened with the numerical values excluded. We feel that signalizing that in the abstract that PFS analysis was the part of the study is beneficial for the scholars assessing the abstract therefore the sentence mentioning PFS was not excluded completely. We hope the Editor will find the compromise acceptable.
- Line 134: The study was approved by the Bioethical Committee (23.02.2022 decision no 134 1072.61201.41.2022) with informed consent waived, as no direct patient participation was planned. Is it OK for the journal?
- In the Instructions for Authors there is a passage allowing for consent waivers and we have successfully published similarly constructed studies in this journal before. Hope this clarifies the issue.
- Line 242-244: Please define abbreviations BMI and BSA when firstly used in the text.
Line 258: Please define abbreviations SOD when firstly used in the text.- Thank you for bringing this shortcoming to our attention. The BMI was defined in line 246 (in the revised version of the manuscript), the BSA in line 248, and the SOD had already been defined in paragraph 2.5 (line 227 in the revised manuscript).
- Significant differences in BMI between LCC and RCC are shown in Table 1, but it is not stated in the text.
- Added the p-value to the passage regarding BMI (line 288 in the revised manuscript).
- Line 295-296: Locoregional recurrence of the primary tumour was observed more frequently in LCC than RCC patients (36.0% vs 296 22.0%). Are these differences significant? Please specify.
- Thank you for pointing that out. The relevant p-value was added (line 302 in the revised manuscript).
- The differences between “Overall Response Rate” and “Overall Response Rate by Investigator” are not clearly stated in the text. Please include a brief explanation or omit one of them.
- The difference had been defined in section 2.5 but indeed nowhere near where the data is presented, which negatively impacts the clarity of the text. Thank you for bringing that to our attention. We have added the short definition and reference to section 2.5 in the footer of the table 4 and changed it’s caption. We have changed the figure 2 to include the other ORR assessment methodology and reworked it’s caption similarly. We have added the passage justifying this dual ORR assessment methodology to e section 2.5 (lines 222-225).
- Line 315-318: Statistical analysis with p-values should be mentioned in the text when significant.
- These values are not available at the moment. I am working with our statisticians to include them in the final manuscript.
- Figure 2 about depth of response is missing.
- A fluke in text formatting. Thank you for noticing. The Figure was re-added (now as Fig. 3 as the other reviewer called for an additional figure) .
- Line 342-343: “All other previously published studies are scarce and single-centre”. Please include these literature references and compare/discuss their results with yours.
- The sentence (lines 364-365 in the revised manuscript) has been rephrased. At the time of manuscript preparation we were aware (through unpublished sources) of one similar project being undertaken but we struggled to find the publication. We have since confirmed with the authors of the study in question that it has not been published yet – hance the change of statement. Additionally a passage on the study being the first source of national RWD data was added to the conclusions (lines 432 – 433).
- The authors should expand the limitations in Discussion section: selection bias, residual confounding (as relevant covariates like different treatments are not assessed), results extrapolation to other populations might be compromised, etc.
- Thank you for this valuable insight. The limitation section has been expanded to address the issues (lines 410-422 in the revised manuscript).
- Response to the treatment could be influenced by body mass index instead of the sidedness of the tumour? In line 281, authors state that “LCC patients had significantly higher BMI than RCC patients”. It would be interesting to include your thoughts about that in Discussion section.
- That is a very interesting insight. Thank you. As I understand the current consensus is that the difference in terms of prognosis between RCC and LCC results from the more aggressive biology and greater resistance to the available treatment options of the former. We have not thought about investigating the cachexia as a potential contributing mechanism but it is an interesting hypothesis now that you have mentioned it. Unfortunately I think we are lacking the data to investigate it in the current publication. My idea is that it would require a proper multivariate analysis which in turn would require the data that are still in complete in our database (ie treatment received or treatment intensity ). This is the scope of one of the forthcoming publications and we hope we could address the issue in full there. Therefore we have decided to only add a mention of this hypothesis in the discussion (lines 379-380 in the revised manuscript). We hope the Editor will find this compromise acceptable.
- Line 401: “with a trend towards improvement in PFS”. Please avoid to state non significant differences in Conclusion section.
- The mention of PFS has been excluded from the Conclusions section. A passage on the study being the first source of national RWD data was added instead (lines 432 – 433).
We hope you will find the changes to the manuscript acceptable.
Thank you again.
Sincerely,
The Authors.